# “Peculiar” Snoring in a 40-Year-Old Patient: A Case Report and Review of Literature

**DOI:** 10.3390/healthcare10061051

**Published:** 2022-06-06

**Authors:** Nicholas-Tiberio Economou, Kyriaki Cholidou, Anastasios Kallianos, Katja Weiss, Beat Knechtle, Pantelis T. Nikolaidis, Georgia Trakada

**Affiliations:** 1Department of Clinical Therapeutics, Division of Pulmonology, National and Kapodistrian University of Athens School of Medicine, Alexandra Hospital, 11528 Athens, Greece; nt_economou@yahoo.it (N.-T.E.); kg.cholidou@yahoo.gr (K.C.); kallianos.pulm@gmail.com (A.K.); gtrakada@hotmail.com (G.T.); 2Institute of Primary Care, University of Zurich, 8000 Zurich, Switzerland; katja@weiss.co.com; 3School of Health and Caring Sciences, University of West Attica, 12243 Athens, Greece; pademil@hotmail.com

**Keywords:** stridor, rapid eye movement sleep behavior disorder, nocturnal enuresis, sleep, dopamine transporter scan

## Abstract

This is a case description of a patient with previously diagnosed rapid eye movement sleep behavior disorder (RBD) and nocturnal enuresis, who complained about a “peculiar sound” while sleeping and, occasionally, while awake during intensive exercise, for the last three months. A home audio recording, which his wife obtained while he was sleeping, showed a high-pitched sound identified as stridor. Full video polysomnography revealed no apneas or hypopneas. The flow-volume loop yielded an inspiratory plateau, indicating variable extrathoracic obstruction. The upper and lower respiratory system computed tomography did not show any anomalies or mechanical lesions (e.g., masses and scars). Fiberoptic laryngoscopy revealed an abnormal activity of the vocal cord abductor during quiet breathing and inspiration in a supine position, leading to partial obstruction. A positive dopamine transporter scan and the coexistence of stridor, RBD, and nocturnal enuresis were diagnostic for multiple system atrophy. The patient was treated with continuous positive airway pressure as a symptomatic therapy for stridor and levodopa.

## 1. Introduction

Stridor is defined as a strained, high-pitched, harsh respiratory sound, mainly inspiratory, occurring only during sleep or during both sleep and wakefulness, and caused by laryngeal dysfunction, leading to narrowing of the rima glottidis (ICD 10 code R06.1). Stridor is a high-pitched inspiratory sound that indicates a dynamic or stable narrowing of airway diameter and is a common symptom encountered by pulmonary physicians, especially when accompanied by dyspnea [1]. When it is recognized, pulmonary function testing (PFTs) and other interventional procedures, such as bronchoscopy, should be performed. The clinical diagnosis of stridor is challenging because patients, family members, or caregivers may be unaware of its presence, especially when it occurs at night. 

Nocturnal stridor can be the initial manifestation of multiple system atrophy (MSA) in 4–5.2% of patients [2,3]. MSA is a progressive, fatal neurodegenerative disorder characterized by a variable combination of autonomic failure, cerebellar ataxia, and parkinsonian feature, with two phenotypes—parkinsonian (MSA-P) and cerebellar (MSA-C) (ICD 10 code G90.3) [4]. MSA is characterized by gross abnormalities of the striatonigral and/or olivopontocerebellar systems, which upon microscopic examination are associated with severe neuronal loss, gliosis, myelin pallor, and axonal degeneration. The term “peculiar snoring” was first introduced by Kakitsuba et al. [5] to describe nocturnal stridor in MSA due to the vibration of the vocal folds in inspiration, with a fundamental acoustic frequency of 260–330 Hz. This sound is different from ordinary snoring, due to the soft palate vibration. Through sound analyses, it can be classified into rhythmic and semi-rhythmic types [6].

Sleep disorders, including sleep fragmentation, insomnia, rapid eye movement sleep behavior disorder (RBD), central sleep apnea (CSA), obstructive sleep apnea (OSA), and nocturnal stridor, are very common in MSA, affecting up to 70% of patients [7]. However, nocturnal stridor, classified as a sleep-related laryngospasm, is the most life-threatening sleeping disorder, with a poor prognostic factor for survival [6,8]. The prevalence of nocturnal stridor in MSA ranges from 12 to 42%, according to different studies [9]. It is considered a diagnostic criterion of possible MSA because of its high diagnostic predictive value [10]. Continuous positive airways pressure (CPAP) and tracheotomy are the treatments of choice for symptomatic control of stridor [9]. We describe a case of a 40-year-old man who initially presented with stridor and was diagnosed with MSA. 

## 2. Case History

A 40-year-old, non-smoking man with an athletic disposition was evaluated in our outpatient Sleep Unit, in the Department of Clinical Therapeutics, National and Kapodistrian University of Athens School of Medicine, at “Alexandra” Hospital, for a “peculiar sound” while sleeping and, occasionally, while being awake, during intensive exercise. This sound differed from ordinary snoring. Inspiratory stridor was discovered based on a home audio recording, which his wife obtained while he was sleeping. 

He was diagnosed with rapid eye movement sleep behavior disorder or REM behavior disorder (RBD) (ICD10:G47.52) and sporadic nocturnal enuresis (ICD10:N39.44) three years ago. Initially, he received clonazepam, which was changed to melatonin. Clonazepam 0.5–1.5 mg was initially introduced with very good results regarding the RBD episodes, but it was soon switched into melatonin 2–4 mg (with results not as good but quite acceptable on RBD) due to nocturnal enuresis deterioration. We hypothesized that clonazepam worsened enuresis due to either its muscle relaxant action or sleep deepening; thus, after having a thorough consultation with the patient, melatonin was chosen as an alternative option. Regarding melatonin, 4 mg/day did ameliorate the RBD symptoms from a clinical point of view (we did not perform another Video-PSG study), but sedation the day after was notable; thus, the patient opted for 2–3 mg of melatonin before bedtime, with more caution on sleep hygiene rules. We failed to increase melatonin beyond 4 mg due to sedation action the day after.

The physical examination was normal and did not show any anomalies. Body mass index (BMI) was 23.4 kgr/m^2^, and oral cavity examination revealed a Mallampati score of 1.

The patient underwent a type I full polysomnography according to American Academy Sleep Medicine (AASM) guidelines to rule out sleep apnea and snoring. RBD episodes have been recorded by Video-PSG in all three REM periods. The variety of the movements went from simple brisk movements/jerks to elaborated movements; in fact, the patient, who is an officer of the Hellenic Navy, used to also be a football (soccer) referee. Therefore, in the most elaborated RBD episode, the patient showed clearly that he was trying to take a yellow/red card out of his pocket in order to penalize a football player (the patient recalled the dream after awakening). As the patient had heavy professional duties (officer in the Hellenic Navy and at the same time an MSci student), he was severely sleep-deprived; all the above-mentioned clues may explain the gross amount of N3/REM sleep (partly due to rebound) and the absence of sleep fragmentation/sleep stage shift. The PSG showed an apnea–hypopnea index of 0.5/h, without periodic leg movements (PLMs) (Figure 1, Table 1). The snore index was 3.2% of the total sleeping duration. Therefore, we concluded that his symptoms were not related to sleep apnea. 

Spirometry was performed, and the flow–volume loop yielded a reduced inspiratory flow during forced inspiration, with a normal expiratory flow pattern (“inspiratory plateau”, Figure 2), indicating a variable extrathoracic airway obstruction. The upper and lower respiratory system computed tomography did not show any anomalies or mechanical lesions (e.g., masses and scars). Fiberoptic laryngoscopy revealed partially abnormal activity of laryngeal muscles during quiet breathing and inspiration, leading to a partial obstruction. The combination of the three symptoms (RBD, nocturnal enuresis, and stridor) and a positive dopamine transporter (DaT) scan was diagnostic for multiple system atrophy (MSA). A DaT scan is a tool used to confirm the diagnosis of Parkinson’s disease. It is a specific type of single-photon emission computed tomography (SPECT) imaging technique that helps visualize dopamine transporter levels in the brain. Levodopa + benserazide hydrochloride tb (200 + 50) mg was introduced at the initial dosage of 50 + 12.5 mg three times/day mostly for slight left arm tremor and bradykinesia. After several months, the dosage was increased to 100 + 25 mg three times/day, with better results. We also prescribed auto-positive airway pressure (APAP, 8–12 cm H_2_O) for symptomatic control of stridor and levodopa.

## 3. Discussion

We present this case report of a 40-year-old man with stridor while sleeping to highlight the diagnostic value of this particular symptom in MSA. This symptom is usually underestimated and underrecognized, especially in young patients, in the differential diagnosis of sleep-related breathing disorders and stridor associated with MSA. 

Stridor is a high-pitched inspiratory sound that indicates a dynamic or stable narrowing of the airway diameter and is a symptom that pulmonary physicians often encounter [1]. When recognized, pulmonary function testing (PFT), imaging studies, and other interventional procedures, such as bronchoscopy, should always be performed. From the pulmonologist’s and sleep expert’s perspective, particular interest can be found in patients who exhibit stridor mainly—but not exclusively—during sleep and is caused by laryngeal neuromuscular dysfunction rather than classical pulmonary causes (e.g., extrathoracic tumors, dynamic airway collapses, etc.) [9].

The prevalence of stridor in MSA ranges from 12% to 42% [9]. The MSA-related nocturnal stridor generates a loud and high-pitched sound during the inspiratory phase, with a fundamental acoustic frequency of 260–330 Hz, which is different from that of ordinary snoring attributed to velopharyngeal space obstruction in patients with obstructive sleep apnea syndrome (OSAS). It was initially described as “peculiar snoring” [5]. Sound analysis can classify it into two types based on acoustic rhythmicity—rhythmic and semi-rhythmic [6].

Nocturnal stridor can be a solitary manifestation in the early stages of MSA-related laryngeal dysfunction—as the function of the vocal cord is only slightly impaired and restricted dilatation of the glottis is not apparent during wakefulness [2]. Currently, two underlying pathogenic mechanisms have been proposed regarding laryngeal stridor: the dystonia hypothesis and the abductor muscle weakness hypothesis [10]. In MSA patients during NREM sleep, dystonic activities of respiratory muscles (intercostal and diaphragmatic) have been reported to coincide with persistent tonic activity and pronounced phasic activation of the adductor laryngeal muscles [11]. This dysfunction of the inhibitory brainstem autonomic pathways may augment the protective airway reflex leading to stridor [12]. The alternative hypothesis states that hypoactivity of the posterior cricoarytenoid muscle due to a lesion of the nucleus ambiguous may result in stridor [13]. Serotonergic neurotransmission in neural sites that project to the nucleus ambiguus is functionally related to the glottic opening.

Stridor is considered a possible indicator for MSA, showing a high diagnostic positive predictive value [14]. Overall, sleep disorders such as sleep fragmentation (52.5%), vocalization (60%), REM sleep behavior disorder (47.5%), and nocturnal stridor (19%) are very common in MSA patients [7]. Our patient had only nocturnal symptoms, RBD, nocturnal enuresis, and stridor. 

The prognostic value of stridor is still under evaluation due to methodologically diverse studies and the lack of homogenization of prognostic factors previously investigated [15]. In the largest study with video polysomnography (VPSG), early onset of stridor (within 3 years from the motoric or autonomic symptom onset) was an independent predictor of a shorter survival rate [3]. However, identifying stridor could be difficult because patients and caregivers may be unaware of its presence, especially when it occurs while sleeping [9]. Stridor during sleep usually reflects an earlier stage of MSA than stridor during wakefulness [9]. Imitation of stridor by the physician or home audio recording can support the diagnostic process.

Further evaluation includes awake or drug-induced sleep laryngoscopy and VDPS to confirm the inspiratory nature of the sound in relation to expiratory intercostal activation and exclude other sleep breathing disorders [9]. Positive airways pressure (PAP) during sleep is the first-line treatment for symptomatic control of stridor, whereas tracheotomy relieves distressing stridor [9]. The impact of therapy on survival is also uncertain [9].

In summary, we presented this case report to highlight the value of stridor as a specific and early indicator of MSA. It is probably underrecognized by patients and physicians alike and must be suspected by the physician when the patient reports high-pitched breathing sounds. 

## Figures and Tables

**Figure 1 healthcare-10-01051-f001:**
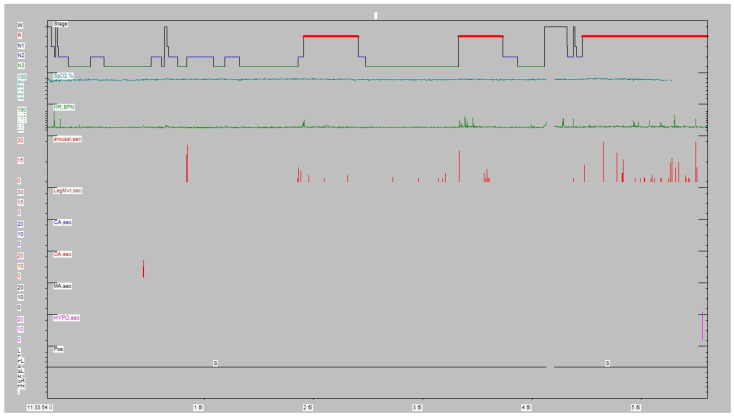
Sleep architecture.

**Figure 2 healthcare-10-01051-f002:**
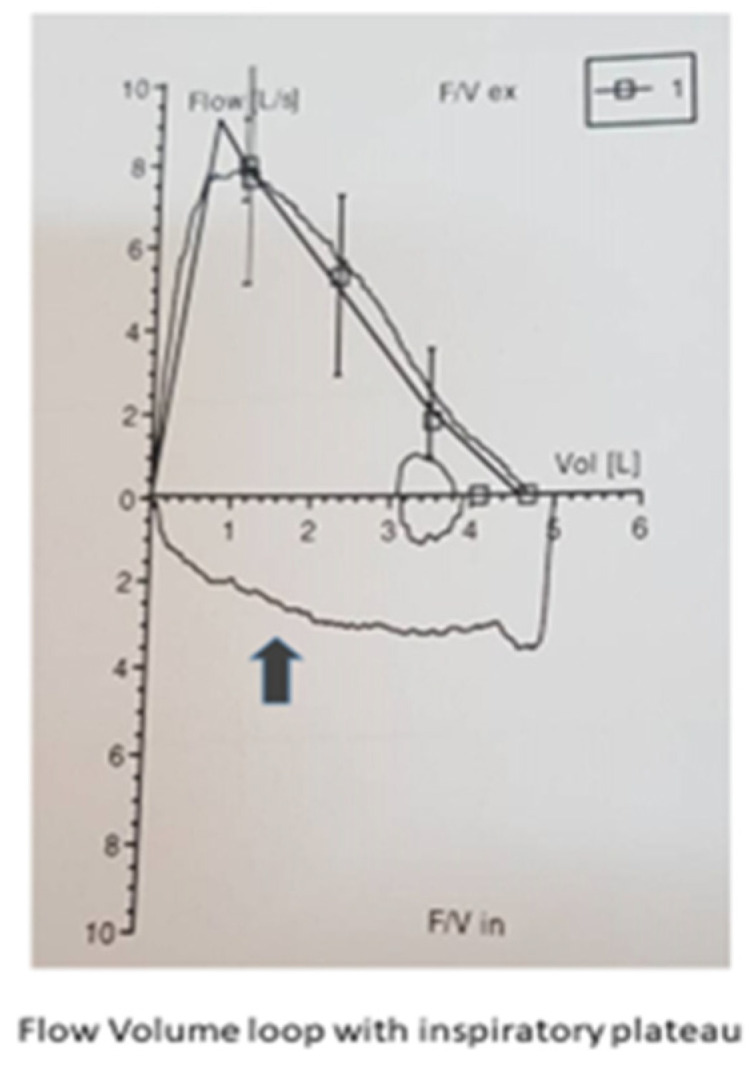
Expiratory flow pattern.

**Table 1 healthcare-10-01051-t001:** Sleep indices.

Sleep Index	Duration
Lights off clock time:	11:33:54 a.m.
Lights on clock time:	5:36:24 p.m.
Total Recording Time (TRT):	372.4 min
Time in Bed (TIB):	362.5 min
Sleep Period Time (SPT):	360.4 min
Total Sleep Time (TST):	344.9 min
Sleep Efficiency:	95.1%
Sleep Onset:	2.1 min
WASO (Wake after Sleep Onset):	15.5 min
REM Latency (from Sleep Onset):	138.5 min

## Data Availability

All data are available by the corresponding author upon reasonable request.

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
