# Peer review of "“Peculiar” Snoring in a 40-Year-Old Patient: A Case Report and Review of Literature"

_healthcare, 2022, doi:10.3390/healthcare10061051_

Round 1

Reviewer 1 Report

1-the manuscript is well written but the authors did not report something new and he stated clearly  that The prevalence of stridor in MSA ranges from 12% to 42%. therefore the manuscript did not add much to the knowledge .

2- why did they give the patient CPAP? the indications are not clear for CPAP therapy,There is no indication for it

Author Response

Reviewer 1

1-the manuscript is well written but the authors did not report something new and he stated clearly that The prevalence of stridor in MSA ranges from 12% to 42%. therefore the manuscript did not add much to the knowledge .

Answer: We agree with the expert reviewer about the need to clarify the novelty of the study. Actually, the aim of this manuscript was to highlight that stridor in an otherwise “asymptomatic” young patient can be an early symptom of MSA, even in the absence of other symptoms.

2- why did they give the patient CPAP? the indications are not clear for CPAP therapy,There is no indication for it

Answer: We agree with the expert reviewer about the need to clarify these aspects: Two main options have been suggested for treating stridor: tracheostomy or CPAP. Although CPAP is not strictly indicated, current literature suggests possible advantages in terms of quality of life and survival, esp. in mild and moderate intensity stridor.

Reviewer 2 Report

Lack of diagnostic criteria of nocturnal stridor.
line 44: It should be "sleep disorders" not "sleeping disorders"
line 62: "Rapid Eye Movement sleep behavior disorder or REM behaviour disorder (RBD) (ICD10:G47.52)" are two same disorders. I do not understand why the authors put them twice?
line 67: please put information about the dosage of clonazepam and melatonin. Any data about the reduction of episodes RBD?
line 65: first polysomnography was undergone after diagnosis RBD? Was the patient free of medications (clonazepam or melatonin)? 
No information about details of Polysomnography procedures, the figure 2. of the hypnogram should be good to present normal sleep architecture. 
Are there data only about respiratory events? 
No REM without atonia episodes? 
No RBD episodes?
Was sleep fragmentation not observed? How can you explain  WASO155 minutes?
The percent of N3 sleep is 43.1%, and REM 35.5%. Is this correct? If Yes, it should be commented! This is not normal sleep architecture!
line 77. The diagnostic criteria of MSA should be present.
line 79. Please describe the parameters of CPAP and the dosage of levodopa. Lack of information about the effectiveness of treatments.
line 106. Should be NREM not nREM
The discussion should be more detailed.

Author Response

Reviewer 2

Lack of diagnostic criteria of nocturnal stridor.

Answer: Stridor is defined as a strained, high-pitched, harsh respiratory sound, mainly inspiratory, occurring only during sleep or during both sleep and wakefulness, and caused by laryngeal dysfunction leading to narrowing of the rima glottides, with icd 10 code R06.1. We added a sentence in Introduction.

line 44: It should be "sleep disorders" not "sleeping disorders"

Answer: We corrected it.

line 62: "Rapid Eye Movement sleep behavior disorder or REM behaviour disorder (RBD) (ICD10:G47.52)" are two same disorders. I do not understand why the authors put them twice?

Answer: We corrected it.

line 67: please put information about the dosage of clonazepam and melatonin. Any data about the reduction of episodes RBD?

Answer: Actually, Clonazepam 0.5-1.5 mg was initially introduced with very good results regarding the RBD episodes, but it was soon switched into Melatonin 2-4 mg (with less good but quite acceptable results on RBD) due to nocturnal enuresis deterioration. We hypothesized that Clonazepam worsened enuresis either due to its muscle relaxant action or due to sleep deepening; thus, after having thorough consultation with the patient Melatonin was chosen as an alternative option. We failed to increase Melatonin beyond 4 mg due to sedation action the day after. We added the information in Case History.

line 65: first polysomnography was undergone after diagnosis RBD? Was the patient free of medications (clonazepam or melatonin)? 

Answer: Yes the patient was drug naïve.

No information about details of Polysomnography procedures, the figure 2. of the hypnogram should be good to present normal sleep architecture.

Answer: Polysomnography was performed according to AASM 2007 guidelines. Figure 1 (we suppose the reviewer refers to this figure) has been replaced with a clearer one; regarding the architecture, actually it was very good with plenty of N3 mostly due to the patient’s chronic sleep deprivation (thus N3 rebound), due to his young age and to the absence of major sleep disorders (OSAS, PLM etc).

Are there data only about respiratory events? 
Answer: The pt had no respiratory events, nor PLMs.

No REM without atonia episodes?

Answer: The patient had 3 NREM/REM cycles. In all REM episodes had both RBD and REM sleep without atonia episodes.

No RBD episodes?

Answer: RBD episodes as stated before, have been recorded by Video-PSG in all 3 REM periods. The variety of the movements went from simple brisk movements/jerks, up to elaborated movements; actually, the patient who is an officer of the Hellenic Navy used to be also a football (soccer) referee. So, in the most elaborated RBD episode, the patient shows clearly that he is trying to take a yellow/red card out of his pocket in order to penalize a football player (the patient recalled the dream after awakening)

Was sleep fragmentation not observed? How can you explain WASO155 minutes?
Answer: As stated before the patient, due to heavy professional duties (officer in the Hellenic Navy and in the meantime MSci student) was severely sleep deprived; moreover, no major sleep disorders (except RBD episodes) have been diagnosed and last he was otherwise a healthy sportive 40 ys old man. All the above-mentioned clues may explain the gross amount of N3/REM sleep (partly due to rebound) and the absence of sleep fragmentation/sleep stage shift.

The percent of N3 sleep is 43.1%, and REM 35.5%. Is this correct? If Yes, it should be commented! This is not normal sleep architecture!
Answer: We agree with the expert reviewer and already addressed this aspect.

line 77. The diagnostic criteria of MSA should be present.
Answer: The diagnostic criteria are presented in Introduction.

line 79. Please describe the parameters of CPAP and the dosage of levodopa. Lack of information about the effectiveness of treatments.

Answer: CPAP parameters were 8-12 cmH2)

line 106. Should be NREM not nREM
Answer: We agree with the expert reviewer and corrected it.

Reviewer 3 Report

I have read the manuscript entitled  “Peculiar” snoring in a 40 years old patient: A case report and review of literature” (Manuscript ID healthcare-1677224) with great interest. It is a case report on a case description of a patient with previously diagnosed Rapid Eye Movement Sleep Behavior Disorder (RBD) and nocturnal enuresis, who complained about a “peculiar sound” while sleeping, finally diagnosed with Multiple System Atrophy.

I have some suggestions to improve manuscript’s eligibility:

Title- I’m not sure if title with „review of literature” is correct. In fact, you don’t provide literaturę review, the discussion section is written in standard form.

Introduction section

  1. Sentence in lines 28-29- stridor is associated with narrowing of upper or lower airway tract? Be more specific, please.
  2. MSA description- lines 36-38- please describe the etiology and patophysiology of MSA in 2-3 sentences

Case history:

  1. Lines 64-65. Physical examination- what did it show? Did you measure weight, height, BMI? Did you conducted oral cavity examination with assessment with Mallampati scale?  Please report detailed physical examination.
  2. Did you perform any laboratory tests?
  3. According to polysomnography assessment. Was it assessed by qualified physician/ polysomnography technician? Did you followed AASM guidelines, or any different guidelines?
  4. Figures 1 and 2 are illegible. Did you provide figures in a single zip archive and at a sufficiently high resolution (minimum 1000 pixels width/height, or a resolution of 300 dpi or higher)?
  5. Please provide snore index.
  6. Line78. Please provide information what DaT scan is.
  7. Lines 79-80. Please provide information about guidelines on CPAP therapy in patients with stridor associated with MSA. Why did you decided to prescribe CPAP? Provide information about CPAP parameters and setting.

Conclusion:

What is take- home message for readers in context of differental diagnosis of sleep- related breathing disorders and stridor associated with MSA ? Please develop in discussion and write a one sentence in conclusions section.

Author Response

Reviewer 3

I have read the manuscript entitled  “Peculiar” snoring in a 40 years old patient: A case report and review of literature” (Manuscript ID healthcare-1677224) with great interest. It is a case report on a case description of a patient with previously diagnosed Rapid Eye Movement Sleep Behavior Disorder (RBD) and nocturnal enuresis, who complained about a “peculiar sound” while sleeping, finally diagnosed with Multiple System Atrophy.

I have some suggestions to improve manuscript’s eligibility:

Title- I’m not sure if title with „review of literature” is correct. In fact, you don’t provide literaturę review, the discussion section is written in standard form.

Answer: We provided all existing data about stridor in MSA.

Introduction section

  1. Sentence in lines 28-29- stridor is associated with narrowing of upper or lower airway tract? Be more specific, please.

Answer: We agree with the expert reviewer and clarified this aspect.

  1. MSA description- lines 36-38- please describe the etiology and patophysiology of MSA in 2-3 sentences

Answer: We agree with the expert reviewer and added this information within text.

Case history:

  1. Lines 64-65. Physical examination- what did it show? Did you measure weight, height, BMI? Did you conducted oral cavity examination with assessment with Mallampati scale?  Please report detailed physical examination.

Answer: Physical examination was normal and oral cavity examination revealed a Mallampati score 1.

  1. Did you perform any laboratory tests?

Answer: Several blood tests were normal. We also performed spirometry.

  1. According to polysomnography assessment. Was it assessed by qualified physician/ polysomnography technician? Did you followed AASM guidelines, or any different guidelines?

Answer: We followed AASM guidelines,

  1. Figures 1 and 2 are illegible. Did you provide figures in a single zip archive and at a sufficiently high resolution (minimum 1000 pixels width/height, or a resolution of 300 dpi or higher)?

Answer: We corrected it.

  1. Please provide snore index.

Answer: We agree with the expert reviewer and added this information (it was 3.2%)

  1. Line78. Please provide information what DaT scan is.

Answer: We agree with the expert reviewer and added this information in the text.

  1. Lines 79-80. Please provide information about guidelines on CPAP therapy in patients with stridor associated with MSA. Why did you decided to prescribe CPAP? Provide information about CPAP parameters and setting.

Answer: We previously answered this issue.

Conclusion: What is take- home message for readers in context of differental diagnosis of sleep- related breathing disorders and stridor associated with MSA ? Please develop in discussion and write a one sentence in conclusions section.

Answer: We agree with the expert reviewer and added this information in the discussion/conclusions.

Round 2

Reviewer 2 Report

 Accept in present form

Author Response

no further changes are required

Reviewer 3 Report

I accept manuscript in current form.

Author Response

no further changes are required